# NBM-BMX, an HDAC8 Inhibitor, Overcomes Temozolomide Resistance in Glioblastoma Multiforme by Downregulating the β-Catenin/c-Myc/SOX2 Pathway and Upregulating p53-Mediated MGMT Inhibition

**DOI:** 10.3390/ijms22115907

**Published:** 2021-05-31

**Authors:** Cheng-Yu Tsai, Huey-Jiun Ko, Shean-Jaw Chiou, Yu-Ling Lai, Chia-Chung Hou, Tehseen Javaria, Zi-Yi Huang, Tai-Shan Cheng, Tsung-I Hsu, Jian-Ying Chuang, Aij-Lie Kwan, Tsung-Hsien Chuang, Chi-Ying F. Huang, Joon-Khim Loh, Yi-Ren Hong

**Affiliations:** 1Ph.D. Program in Environmental and Occupational Medicine, College of Medicine, Kaohsiung Medical University and National Health Research Institutes, Kaohsiung 807, Taiwan; moutzyy691010@yahoo.com.tw (C.-Y.T.); aijliekwan@yahoo.com.tw (A.-L.K.); thchuang@nhri.edu.tw (T.-H.C.); 2Department of Neurosurgery, Kaohsiung Medical University Hospital, Kaohsiung 807, Taiwan; 3Graduate Institute of Medicine, College of Medicine, Kaohsiung Medical University, Kaohsiung 807, Taiwan; o870391@yahoo.com.tw (H.-J.K.); 4a1h0010@gmail.com (Y.-L.L.); 4Department of Biochemistry, College of Medicine, Kaohsiung Medical University, Kaohsiung 807, Taiwan; sheanjaw@kmu.edu.tw; 5New Drug Research & Development Center, NatureWise Biotech & Medicals Corporation, Taipei 112, Taiwan; alison.hou@naturewise.com.tw; 6Institute of Biopharmaceutical Sciences, National Yang Ming Chiao Tung University, Taipei 112, Taiwan; jia.tehseen@gmail.com (T.J.); mountain1002@yahoo.com.tw (T.-S.C.); 7Program in Molecular Medicine, National Yang Ming Chiao Tung University, Taipei 112, Taiwan; Laduree120@ym.edu.tw; 8Ph.D. Program for Neural Regenerative Medicine, College of Medical Science and Technology, Taipei Medical University, Taipei 115, Taiwan; dabiemhsu@tmu.edu.tw (T.-I.H.); chuangcy@tmu.edu.tw (J.-Y.C.); 9Immunology Research Center, National Health Research Institutes, Miaoli 350, Taiwan; 10Department of Biological Sciences, National Sun Yat-Sen University, Kaohsiung 804, Taiwan; 11Department of Medical Research, Kaohsiung Medical University Hospital, Kaohsiung 807, Taiwan

**Keywords:** HDAC8, GBM, TMZ, MGMT, connectivity map, β-catenin, p53

## Abstract

Although histone deacetylase 8 (HDAC8) plays a role in glioblastoma multiforme (GBM), whether its inhibition facilitates the treatment of temozolomide (TMZ)-resistant GBM (GBM-R) remains unclear. By assessing the gene expression profiles from short hairpin RNA of HDAC8 in the new version of Connectivity Map (CLUE) and cells treated by NBM-BMX (BMX)-, an HDAC8 inhibitor, data analysis reveals that the Wnt signaling pathway and apoptosis might be the underlying mechanisms in BMX-elicited treatment. This study evaluated the efficacy of cotreatment with BMX and TMZ in GBM-R cells. We observed that cotreatment with BMX and TMZ could overcome resistance in GBM-R cells and inhibit cell viability, markedly inhibit cell proliferation, and then induce cell cycle arrest and apoptosis. In addition, the expression level of β-catenin was reversed by proteasome inhibitor via the β-catenin/ GSK3β signaling pathway to reduce the expression level of c-Myc and cyclin D1 in GBM-R cells. BMX and TMZ cotreatment also upregulated WT-p53 mediated MGMT inhibition, thereby triggering the activation of caspase-3 and eventually leading to apoptosis in GBM-R cells. Moreover, BMX and TMZ attenuated the expression of CD133, CD44, and SOX2 in GBM-R cells. In conclusion, BMX overcomes TMZ resistance by enhancing TMZ-mediated cytotoxic effect by downregulating the β-catenin/c-Myc/SOX2 signaling pathway and upregulating WT-p53 mediated MGMT inhibition. These findings indicate a promising drug combination for precision personal treating of TMZ-resistant WT-p53 GBM cells.

## 1. Introduction

Glioblastoma multiforme (GBM) is one of the most malignant tumors, and it has an aggressive pattern and a high recurrence rate; it is a World Health Organization grade IV astrocytoma [1]. Despite multimodalities treatment with surgery and concomitant radiation and chemotherapy, patients with GBM still have a poor prognosis, with a mean survival of <15 months, which indicates therapeutic resistance [2,3,4]. Traditional chemotherapy for GBM includes temozolomide (TMZ), which is an oral alkylating agent that induces cell cycle arrest at the G2/M phase, leading to apoptosis [5,6]. However, less than 50% of patients respond to TMZ due to the overexpression of O6-methylguanine methyltransferase (MGMT), which reverses the methylation of the O6 position of guanine, thereby repairing DNA in GBM cells and resisting the chemotherapeutic effect [7,8]. In addition to promoter methylation, MGMT is regulated by various transcription factors, such as p53, Sp1, NF-κB, CEBP, and AP-18. Among these, p53 downregulates MGMT transcription by directly interacting with the MGMT promoter [9,10]. Thus, in addition to MGMT promoter methylation, p53 could regulate MGMT expression and cause TMZ resistance. Therefore, additional mechanisms regulating MGMT must be identified to overcome TMZ resistance.

Epigenetic modulation, such as DNA methylation, histone modifications, and microRNA (miRNA) modifications, plays a critical role in many cancers. Histone deacetylases (HDACs) are essential in tumorigenesis and development through epigenetic regulation [11]. Abnormal HDAC overexpression is observed in many tumors, such as colorectal cancer [12], breast cancer [13], and leukemia [14]. HDAC inhibitors (HDACis) are promising therapies for several types of cancer, such as breast cancer [15], colon cancer [16], hepatocellular carcinoma [17], and medulloblastoma [18], because they increase the susceptibility of tumor cells to apoptosis. HDACs can interact with p53 through deacetylation and reduce its transcriptional activity [19]. Accordingly, HDACis can increase p53 acetylation and p53-dependent activation of apoptosis and senescence. Moreover, HDACs are related to the cancer stem cells (CSCs) of GBM (GSCs), and HDACis can reduce the GSC population [20]. HDAC knockdown inhibits cell proliferation and impairs GSC activity, which can be useful for treating TMZ-resistant GBM (GBM-R). HDAC families are classified into four groups by their sequence identity and catalytic activity: class I (HDAC1, 2, 3, and 8), class II (HDAC4, 5, 6, 7, 9, and 10), class III (SIRT1-7), and class IV (HDAC11) [21]. Among all of the HDAC family, HDAC8 is a unique 42-kDa protein with 377 amino acids [22] and is particularly abundant in the brain, prostate, and kidneys [23]. It can localize to either the nucleus (primary site) or the cytoplasm [24]. It serves as an oncogene in various tumors, including gastric cancer, neuroblastoma, T-cell lymphocytes, hepatocellular carcinoma, and breast cancer. HDAC8 downregulation increases sensitivity to chemotoxicity in neuroblastoma [25] and induces the differentiation of malignant cells into neurons [26]. In pharmacological and clinical view, pan-HDACis and isoform-selective HDACis have been used in many clinical trials for cancer treatments. Several isoform-selective HDACis, including HDAC8, have shown promising results in GBM preclinical trials due to their minimal toxicity compared with pan-HDACis [27,28,29,30]. In another learning and memory study, NBM-BMX (BMX), a novel small-molecule isoform-selective HDAC8 inhibitor, has the lowest toxicity and the ability to cross the blood–brain barrier [31]. However, BMX has not been extensively studied in TMZ-resistant GBM (GBM-R).

The present study investigated whether BMX can enhance the TMZ-mediated cytotoxic effect on GBM-R cell lines. Our results suggest that HDAC8 inhibition (BMX) could overcome TMZ resistance in GBM-R cells by enhancing the TMZ-mediated cytotoxic effect by downregulating the β-catenin/c-Myc/SOX2 signaling pathway and upregulating p53-mediated MGMT inhibition.

## 2. Results

### 2.1. Pathway Analysis for Potential Expression Profile of HDAC8 Inhibitor through Bioinformatics Tools

To explore possible mechanisms of HDAC8 inhibitor and gene involvement, we used Connectivity Map (C-Map), the Library of Integrated Network-Based Cellular Signatures Unified Environment (CLUE) systemic database (https://clue.io/, accessed on 10 May 2021), and the ConsensusPathDB (CPDB) platform (http://cpdb.molgen.mpg.de/, accessed on 10 May 2021) for the comprehensive mechanism analysis. We utilized two bioinformatics processes, direct and indirect analyses, respectively (Figure 1A). For the direct analysis, HepG2 cells were treated with BMX in a L1000 plate, which responded to the biological function of BMX (Figure 1A, right). The significant differentially expressed genes (1583 up-regulation and 900 down-regulation) with 1.5-fold change were used to query the CPDB platform to reveal the potential pathways (*p*-value < 0.05). Next, we analyzed the HDAC8 inhibition function through an indirective approach, the pattern matching algorithm of the CLUE platform. Using shRNA *HDAC8* signature as the simulation of BMX treatment (HDAC8 inhibitor), we then accessed CLUE, which computed over 1 million profiles to match the similar signature-pattern from 19,811 small molecule compounds or gene perturbations (e.g., 18,493 shRNAs, 3462 over-expression constructs), and then obtained the connectivity score. The positive score denoted a similar mechanism between query and instance signatures, while the negative meant the opposite function. Our criteria were selected above 90 connectivity scores of compounds (CPs), knockdown genes (KDs), overexpression genes (OE), and perturbagen classes (PCLs). CLUE clustered the similar function compounds or same family genes into a particular group, which could postulate as the mechanism of action. However, this big data system did not offer detailed pathway information. Thus, we combined the CPDB platform for complementary analysis from shHDAC8 and BMX-treated cells (Figure 1A, left). These different bioinformatics pipelines would obtain several mechanisms/pathways, and we intersected these two datasets to filter the possible potential pathways. The Wnt signaling pathway is one of the top-ranking mechanisms uncovered via our multi-databases platform (Figure 1B).

### 2.2. BMX Enhanced the TMZ-Mediated Cytotoxic Effect to Inhibit the Growth and Proliferation in GBM-R Cells

To investigate whether HDAC8 is correlated with therapy-resistant GBM, we examined the HDAC8 expression level of two parent GBM cell lines (A172 and U87MG, wild-type p53 (WT-p53), Appendix A) and two TMZ-resistant GBM cell lines (A172-R and U87MG-R, variants of WT-p53). HDAC8 overexpression was detected in both GBM-R cell lines (Appendix A).

We used NBM-BMX (provided by Nature Wise Biotech & Medicals Corporation; BMX was used in this manuscript) as an HDAC8 inhibitor to mimic the effect of shRNA HDAC8 for further experiments. The structure of BMX (397.46 Da) is shown in Figure 2A. Although BMX was already identified as an HDAC8 inhibitor in an enzymatic activity study and inhibition assay [31], we verified that BMX is an HDAC8 inhibitor by treating the four cell lines with BMX and detecting BMX-induced inhibition of HDAC8 mRNA and protein expression (Appendix A).

Because HDAC8 may be related to TMZ resistance in GBM, we considered that HDAC8 inhibition might enhance the sensitivity of the TMZ-mediated cytotoxic effect in both GBM and GBM-R cells. To examine whether a combination effect exists between BMX and TMZ in treating GBM and GBM-R, we treated A172/A172-R and U87MG/U87MG-R cells with BMX and TMZ alone and in combination. An MTT assay was performed to assess cell proliferation and cell viability for the BMX-alone, TMZ-alone, and combination groups under different concentrations at 24, 48, and 72 h. In each treatment alone group, the results suggested that the cytotoxic effect in each group increased in a time-dependent manner (Appendix A). The results revealed that the IC_50_ values of BMX alone were 21.00 ± 2.34 μM/ > 52.64 ± 3.62 μM in A172/A172-R cells and 29.84 ± 2.32 μM/ > 68.13 ± 4.69 μM in U87MG/U87MG-R cells (Figure 2B), suggesting that BMX alone could inhibit GBM cell proliferation, but not inhibit GBM-R cell proliferation. In addition, the IC_50_ values of TMZ alone were 73.48 ± 3.65 μM/80.99 ± 1.68 μM in A172/U87MG cells and 595.07 ± 23.42 μM/302.51 ± 15.24 μM in A172-R/U87MG-R cells, confirming the reliability of GBM-R cells (Figure 2C). In the combination treatment group, BMX 10 μM was used to combine with different doses of TMZ (Figure 2D) and TMZ 50 μM (the same as the maintenance concentration in GBM-R cell lines) combined with different concentrations of BMX (Figure 2E) to determine which dose of BMX and TMZ can most enhance the TMZ-mediated cytotoxic effect in GBM-R cells. The data revealed that 50 μM TMZ with 10 μM BMX exerted the highest cytotoxic effect in both GBM-R cell lines. We used this combination in a time-dependent manner and noted a cytotoxic effect in 48 h (Figure 2F, in 48 h BMX10 μM: 0.88×, 0.77×, 0.63×; BMX and TMZ: 0.74×, 0.56×, 0.47×). The clonogenic assay also revealed that 10 μM BMX with 50 μM TMZ, rather than BMX alone, suppressed GBM-R cells (Figure 2G). Collectively, these data suggest that the combination treatment inhibits the growth and proliferation of GBM cells (U87MG and A172) and GBM-R cells (U87MG-R and A172-R) in that the combination of 10 μM BMX and 50 μM TMZ exerts the highest cytotoxic effect (suppression of cell proliferation and cell viability) on GBM-R cells. Nevertheless, the cell viability of BMX alone was still moderately decreased to show the partial ability of pharmacological cytotoxic effect in A172R/U87R, compared with TMZ alone with no suppression effect. Thus, the combined BMX and TMZ treatment was compared with BMX alone in the further experiments.

### 2.3. BMX Enhanced the TMZ-Mediated Cytotoxic Effect by Targeting the Wnt/β-Catenin/GSK3β Pathway in GBM-R Cells

We investigated the mechanism by which BMX enhances the TMZ-mediated cytotoxic effect in GBM-R cells. On the basis of the pathway analysis, we postulated that the canonical Wnt signaling (also known as Wnt/β-catenin) pathway is involved in the proliferation of GBM-R cells. The genetic background for each cell indicated no mutation in Wnt genes such as adenomatous polyposis coli and β-catenin (CTNNB1). Therefore, we examined phospho-β-catenin (Ser33/Ser37/Thr41) as a β-catenin active form for detecting the β-catenin status. GSK3β (S9) was used for β-catenin phosphorylation to degrade β-catenin. The results indicated that 10 μM BMX with 50 μM TMZ reduced the protein levels of β-catenin directly and reduced the protein levels of phospho-β-catenin (Ser33/Ser37/Thr41) through phosphorylation by GSK3β in U87R and A172R cells, whereas BMX alone only slightly reduced these levels. Moreover, the phosphorylation level of GSK3β (S9) also decreased, indicating that GSK3β activity increased and β-catenin was phosphorylated (Figure 3A). We further examined the effects of BMX on the proliferative markers c-Myc and cyclin D1 and noted that BMX both with and without TMZ could decrease their levels (Figure 3B).

To verify that the β-catenin protein level decreased due to protein degradation, the GBM-R cells were treated with the proteasome inhibitor MG132. The results revealed that MG132 application reversed β-catenin degradation and increased c-Myc and cyclin D1 expression under 10 μM BMX and 50 μM TMZ (Figure 3C). These results demonstrated that BMX enhanced GSK3β activity through Ser9 phosphorylation downregulation, which in turn enhanced β-catenin phosphorylation at Ser33/Ser37/Thr41, triggering protein degradation. Taken together, these data revealed that 10 μM BMX and 50 μM TMZ enhanced TMZ-mediated cytotoxic effects, partly via the Wnt/β-catenin/GSK3β pathway, thus reducing GBM-R cell proliferation.

### 2.4. BMX Enhanced the TMZ-Mediated Cytotoxic Effect by Promoting TMZ-Mediated Apoptosis in GBM-R Cells

To investigate whether BMX can induce cell cycle arrest, we analyzed the effects of BMX (5 μM and 10 μM) alone and combined with 50 μM TMZ on the cell cycle in the A172-R and U87MG-R cell lines. The results revealed that 10 μM BMX alone induced cell cycle arrest in the G0/G1 phase in A172-R cells (70.34%) and U87MG-R cells (77.95%). Next, 5 and 10 μM BMX with 50 μM TMZ not only increased the amount of cell cycle arrest in G0/G1 but also caused arrest in the sub-G1 phase (apoptosis) in both GBM-R cell lines (Figure 4A–C).

Flow cytometry revealed that BMX combined with TMZ yielded a high percentage of apoptotic cells in A172-R/U87MG-R cell lines (21.7%/25.95%) in a dose-dependent manner (Figure 4D). Moreover, late apoptosis was also predominant after treatment with 10 μM BMX and 50 μM TMZ (Figure 4E). Thus, BMX alone could only induce cell cycle arrest and suppress cell proliferation but not induce apoptosis, whereas the BMX and TMZ combination could also promote TMZ-mediated apoptosis, leading to enhanced cytotoxicity in GBM-R cells.

### 2.5. BMX Enhanced the TMZ-Mediated Cytotoxic Effects by WT-p53 Mediated MGMT Inhibition in GBM-R Cells

Because the BMX and TMZ combination could promote TMZ-mediated apoptosis, we speculated that BMX might enhance TMZ-mediated apoptosis through WT-p53 mediated MGMT inhibition. First, we examined WT-p53 and MGMT levels in A172/A172-R and U87MG/U87MG-R cells, confirming that TMZ resistance is related to WT-p53 and MGMT (Figure 5A). The bioinformatics analysis also suggested that only 33% of patients have the p53 mutation, and the others have p53 WT (Appendix A). Next, the TCGA and driverDB databases were assessed to check the overall survival rate between p53 mutation and p53 WT. By colony formation assay (Appendix A), it was clearly revealed that p53 WT cases showed poor prognosis in GBM patients as compared to mutated cases.

We examined the proapoptotic signaling system in WT-p53 mediated apoptosis. MGMT was also examined for TMZ repair ability. The results revealed that the levels of proapoptotic makers, such as P21, Bax/Bcl2, and Puma, increased and that those of MGMT decreased after treatment with BMX both without and with 50 μM TMZ. However, cleaved caspase-3 was only noted with the combination of 10 μM BMX and 50 μM TMZ (Figure 5B). To clarify whether apoptosis is induced by BMX alone, TMZ alone, or their combination in WT-p53 mediated MGMT inhibition, we treated A172-R and U87MG-R cells with 50 μM TMZ without or with 5 and 10 μM BMX. TMZ alone could only moderately suppress MGMT expression without increasing WT-p53 and DNA damage marker (WT-p53-ser15). However, MGMT expression obviously decreased with the combination of 10 μM BMX and 50 μM TMZ. Moreover, WT-p53 and DNA damage markers’ (WT-p53-ser15) expression levels also increased, meaning that MGMT was negatively regulated by WT-p53mediated apoptosis (Figure 5C).

Moreover, by assessing scatter plot for p53 WT and mutant cells (Appendix A), it was found that GBM p53 WT cells are MGMT hypermethylated and lower the MGMT mRNA and protein expression as well. In addition, TMZ alone could not induce WT-p53 mediated apoptosis in GBM-R cells. Together, these data suggested that BMX and TMZ combination could enhance the TMZ-mediated cytotoxic effect through WT-p53 mediated MGMT inhibition in GBM-R cells. However, BMX alone could moderately decrease MGMT levels but not induce WT-p53 mediated apoptosis in GBM-R cells.

### 2.6. The BMX and TMZ Combination Treatment Reduced GSC Formation in GBM-R Cells

Because GSC markers are the core of GBM resistance, we examined the levels of GSC markers in all cell lines; high expression levels of CD133, CD44, and SOX2 were detected in A172-R and U87MG-R cells, implying that TMZ resistance is partly related to GSC markers (Figure 6A). Furthermore, treatment with 10 μM BMX and 50 μM TMZ clearly reduced the expression levels of CD133, CD44, and SOX2 in both GBM-R cell lines (Figure 6B). Thus, the BMX and TMZ combination could enhance the TMZ-mediated cytotoxic effect by attenuating GSC markers to convert the stemness phenotype in GBM-R cells.

We also examined HDAC8 and GSC markers in TMZ-resistant GBM human tissues through immunohistochemistry (Figure 6C). The results revealed that HDAC8 and GSC are closely related to TMZ resistance in GBM.

## 3. Discussion

Although preclinical studies have indicated that HDACis have antitumor effects in glioma [28,29,30], their role in the treatment of chemotherapy-resistant GBM is still unclear. This is the first study to demonstrate that BMX, a novel iso-selective HDAC8 inhibitor, can enhance TMZ-mediated cytotoxic effect not only by downregulating the β-catenin/c-Myc/SOX2 pathway to inhibit stemness but also by upregulating WT-p53 mediated MGMT inhibition to induce apoptosis in TMZ-resistant GBM cells. Moreover, we also revealed that the inverse correlation of WT-p53/MGMT reversion and the Wnt/β-catenin/GSKβ signaling pathway may be involved in the oncogenic role in GBM and GBM with TMZ resistance.

On the basis of our findings, we propose the following working model (Figure 7): first, the β-catenin/c-Myc/cyclin D1/SOX2 signaling pathway in TMZ-resistant GBM (pathway on the right side). According to our previous studies and the bioinformatics analysis in this study, the Wnt/β-catenin/GSK3β pathway can influence therapy selection for GBM [32]. Our findings demonstrated that BMX both without and with TMZ (thin and thick lines) could enhance GSK3β activity by downregulating Ser9 phosphorylation, which in turn enhanced β-catenin phosphorylation at Ser33/Ser37/Thr41, triggering β-catenin protein degradation. β-catenin degradation was confirmed with MG132 as a proteasome inhibitor. Undegraded β-catenin translocates into the nucleus to bind to TCL4 and activate downstream target genes, such as c-Myc and cyclin D1, to induce cell proliferation and continue the cell cycle. Both BMX alone (thin lines) and BMX with TMZ (thick lines) suppressed c-Myc and cyclin D1 expression and induced cell cycle arrest. However, BMX alone could not induce cell cycle arrest in the sub-G1 phase. Only the BMX with TMZ combination induced profound cell cycle arrest and proceeded to the sub-G1 phase, meaning that it possible induces late-apoptosis in GBM-R cells (dotted line in the right lower part).

In addition, GSCs play a vital role in therapeutic resistance in GBM. They are characterized by their self-renewal ability, both in vitro and in vivo, through high expressions of neuronal stem cell markers, such as CD133 and CD44, as well as transcription factors, such as SOX2 [33]. Our data also revealed that BMX both without and with TMZ attenuated not only CD133 and CD44 but also SOX2 by downregulating the GSC phenotype to suppress stemness. As previously reported, c-Myc is also required to maintain glioma CSCs [34]. Collectively, we demonstrated that BMX alone and BMX with TMZ suppressed cell proliferation by enhancing the TMZ-mediated cytotoxic effect via the β-catenin/c-Myc/cyclin D1/SOX2 signaling pathway in GBM-R cells.

Next, WT-p53 mediated MGMT inhibition in TMZ-resistant GBM (pathway on the left side). The mechanism of action of TMZ in GBM is methylation of the O6 position to guanine to damage DNA. MGMT reverses methylation to repair DNA in GBM cells and to exert GBM resistance [7]. Although an MGMT-independent pathway also plays a critical role in TMZ resistance [35,36,37], the MGMT-dependent pathway is still considered the major pathway for TMZ resistance. In our study, GBM-R cell lines (A172-R and U87MG-R) expressed high levels of the MGMT protein, thus verifying that the MGMT-dependent pathway is indeed a major mechanism for TMZ resistance in these cell lines. Recent studies have also reported that HDAC8 inhibition could reduce MGMT protein levels [30]. Therefore, we speculated that BMX may suppress MGMT expression in MGMT-dependent GBM-R cell lines to weaken the ability of MGMT for DNA damage repair. In the GBM-R cell lines, BMX alone moderately decreased MGMT expression and TMZ alone did not (Figure 5B,C), but their combination obviously reduced the MGMT protein levels and enhanced TMZ-mediated apoptosis in an MGMT-dependent manner.

WT-p53 is closely correlated with MGMT expression and negatively regulates MGMT transcription [9,10]. TMZ induces DNA damage by increasing WT-p53 expression [38], and HDAC1 inhibition prevents chemotherapeutic resistance in U-87MG cells by WT-p53 reactivation [39]. Moreover, p53 protein and transcripts are decreased by HDAC inhibitors or HDAC8 knockdown [40,41,42]. Taken together, we proposed that inhibition of HDAC maybe decreased MGMT via WT-p53 restoring (all P53 lanes in Figure 5A–C). In our work, we demonstrated that BMX alone (thin lines in the left part) moderately increased WT-p53 level to moderately downregulate MGMT expression, leading to still maintaining the ability of DNA repair (Figure 5B). However, still more evidence is still needed to confirm the direct relationship of inhibition of HDAC8 with MGMT in GBM-R cells (dashed line in middle part). Thus, we speculate that HDAC inhibition may reduce MGMT expression through WT-p53 reactivation. We demonstrated that BMX alone moderately increased the WT-p53 level and moderately downregulated MGMT expression, leading to the maintenance of DNA repair. Moreover, BMX alone also induced the cell cycle arrest marker (P21). However, additional studies are required to elucidate the relationship between HDAC8 inhibition and MGMT expression in GBM-R cells. In the present study, the BMX with TMZ combination (thick lines in left part) induced extensive DNA damage through WT-p53 (and ser15) overexpression and downregulated MGMT expression, eventually leading to WT-p53 mediated apoptosis (Figure 5C). This combination (compared with BMX alone) could also increase the expression of cell cycle arrest marker (P21), proapoptotic proteins (Bax/Bcl2 and Puma), and induce cleaved caspase-3 expression for WT-p53 mediated apoptosis. Taken together, these results implied that BMX (thin lines in the left part) alone only partially induced WT-p53 mediated MGMT inhibition but that the BMX and TMZ combination (thick lines in the left part) enhanced the TMZ cytotoxic effect in GBM-R cell lines to overcome TMZ resistance.

More generally, this work shows the BMX, a novel small-molecule isoform-selective HDAC8 inhibitor, overcomes TMZ resistance by enhancing TMZ-mediated cytotoxic effect by downregulating the β-catenin/c-Myc/SOX2 signaling pathway and upregulating WT-p53 mediated MGMT inhibition. These findings indicate a promising drug combination for precision personal treating of TMZ-resistant WT-p53 GBM cells in the future.

## 4. Materials and Methods

### 4.1. Cell Culture and Reagents

Four GBM cell lines, U87, U87R, A172, and A172R, were used in this study. The American Type Culture Collection (ATCC; Manassas, VA, USA) provided human GBM cell lines U87-MG (ATCC HTB-14; GBM of unknown origin) and A172 (ATCC CRL-1620; ATCC). U87R and A172R cells were obtained from Dr. Tsung-I Hsu and Dr. Jian-Ying Chung (The Ph.D. Program for Neural Regenerative Medicine, College of Medical Science and Technology, Taipei Medical University, Taipei, Taiwan) [42]. These cells were maintained in Dulbecco’s modified Eagle medium (DMEM) with 10% fetal bovine serum (FBS) and 50 μM TMZ for at least 60 days. TMZ resistance in U87R and A172R cells was confirmed using the colony formation assay (Appendix A). Cells were cultured in DMEM supplemented with 10% FBS, 100 U/mL penicillin, and 100 mg/mL streptomycin (all from Gibco; Thermo Fisher Scientific, Waltham, MA, USA) and maintained in a humidified incubator at 37 °C and 5% CO_2_. NBM-BMX (BMX), (*E*)-2-(4-Methoxybenzyloxy)-3-prenyl-4-methoxy-*N*-hydroxycinamide, was provided by NatureWise Biotech & Medicals Corporation (Taipei, Taiwan).

### 4.2. Cell Proliferation and Colony Formation Assays

We plated 3000 GBM cells per well in 96-well plates and allowed them to adhere overnight. To validate cell line responsiveness to BMX and TMZ monotherapy, the cells were treated with different doses of BMX or TMZ for 24, 48, and 72 h. To confirm cell responsiveness to the BMX-TMZ combination, cells were either treated with different doses of TMZ (0–800 µg/mL) with or without BMX (10 μM) for 24, 48, and 72 h or with different doses of BMX (0–50 μM) with or without TMZ (50 μM) for 24, 48, and 72 h. Following treatment, the absorption value was measured using a CCK8 kit (Targetmol, Shanghai, China) at the indicated time points. The results are reported as the mean ± standard deviation of at least three replicates.

A172, A172-R, U87MG, and U87MG-R cells were seeded (1000 cells/dish) into 6-cm culture dishes and incubated for 14 days. The cells were washed three times with phosphate-buffered saline, fixed in 4% paraformaldehyde for 30 min, and stained with 0.1% crystal violet for 20 min at 25 °C. The colonies were carefully washed with tap water, and then the number of colonies, defined as at least 50 cells, were counted and analyzed. The results are expressed as the average colony count ± SE from three independent experiments.

### 4.3. Reverse Transcription-Quantitative Polymerase Chain Reaction (RT-qPCR)

The ABI Prism^®^ 7700 Sequence Detection System (Applied Biosystems, Foster City, CA, USA) was used for quantitative analysis of mRNA expression. The cells (2 × 10^5^) were seeded in 6-well plates, and total RNA was extracted using Tissue Total RNA Mini Kit (Geneaid, Taipei, Taiwan). A 10-ng sample of total RNA was transcribed into cDNA by using a High-Capacity cDNA Reverse Transcription Kit (Applied Biosystems). Gene expression was quantified using Fast SYBR Green Master Mix (Applied Biosystems) following the procedures provided by the manufacturer, with 18s as the inner reference. All procedures were performed according to the manufacturer’s protocols. The thermocycling conditions were as follows: 50 °C for 2 min, 95 °C for 10 min, and 40 cycles of 95 °C for 15 s and 60 °C for 1 s. Each sample was analyzed in triplicate. The threshold cycle (Ct) values were calculated using the StepOnePlus (Applied Biosystems) software. The relative expression of each mRNA was calculated using the 2−(ΔCt) method. The primer sequences for HDAC8 were as follows: HDAC8 forward 5′-GCGTGATTTCCAGCACATAA-3′; HDAC8 reverse 5′-ATACTTGACCGGGGTCATCC-3′. 18 s forward 5′-TCAAGTGCAGTGCAACAACTC-3′; 18 s reverse 5′-AGAGGACAGGGTGGAGTAATCA-3′.

### 4.4. Flow Cytometric Analysis of the DNA Cell Cycle

For the DNA cell cycle, following treatment with different doses of BMX (0–10 μM) in the presence or absence of TMZ (50 μM) for 48 h, the cells were harvested through trypsinization, washed twice with phosphate-buffered saline, and fixed in methanol. The cells were then washed again, subjected to RNase A at a final concentration of 0.05 mg/mL (Sigma-Aldrich; Merck Millipore, Darmstadt, Germany), and incubated with 10 µg/mL propidium iodide (PI; Sigma-Aldrich; Merck Millipore) at 4 °C for 15 min in the dark. Cell cycle analysis was conducted using a fluorescence-activated cell sorting (FACS) flow cytometer (Attune NxT flow cytometer, Thermo Fisher Scientific).

### 4.5. Flow Cytometric Analysis of Apoptosis

To analyze cell apoptosis in different doses of BMX (0–10 μM) in the presence or absence of TMZ (50 μM), FITC-labeled annexin V/PI staining was performed using the CF^®^488A Annexin V and PI Apoptosis Kit (Fremont, CA, USA), according to the manufacturer’s instructions. The analysis by flow cytometry of PI and annexin was performed 48 h post-treatment. A total of 10,000 nuclei were measured using an FACS flow cytometer (Attune NxT flow cytometer, Thermo Fisher Scientific).

### 4.6. Immunohistochemical Staining

Immunohistochemical staining was performed on 4-μm-thick paraffin sections. The sections were dewaxed hydrated and placed at 4 °C overnight. For antibodies against CD133 (AP1802a, Abgent, San Diego, CA, USA), P62 (ab56416, Abcam, Cambridge, MA, USA), and LC3II (AP1802a, Abgent), the standard avidin–biotin complex procedures were employed. After the sections were returned to room temperature, biotinylated secondary antibodies and horseradish-labeled streptavidin were added. The samples were then incubated in an oven at 37 °C. Subsequently, DAB color development, hematoxylin counterstaining, gradient alcohol dehydration, and xylene transparent were conducted. All samples were sealed with neutral gum afterwards. Human brain tissues: the ethics statements in this study were approved by the Institutional Review Board of Kaohsiung Medical University Hospital (No. KMUHIRB-F(I)-20200024). Informed consent was obtained from all subjects involved in the study.

### 4.7. Western Blot Analysis

The cells were collected and lysed in RIPA lysis buffer (EMD Millipore Billerica, MA, USA, 10× RIPA buffer) containing a protease inhibitor. Protein concentration was determined using a protein assay kit (Bio-Rad Laboratories, Hercules, CA, USA). The SDS loading buffer was mixed with the protein samples. Proteins (20 µg/lane) were separated using 8–12% SDS-PAGE and transferred to a PVDF membrane, which was blocked with 5% bovine serum albumin at room temperature for 1 h in tris-buffered saline (TBS)-Tween 20 (0.5%; TBS-T), incubated with primary antibodies overnight at 4 °C, and then incubated with horseradish peroxidase (HRP)-conjugated secondary antibodies at room temperature for 1 h. After washing thoroughly with TBS-T, the HRP signals were detected with a chemical HRP substrate. The antibodies we used are listed in Appendix A. Signal of each target protein was visualized by incubation with an enhanced chemiluminescent reagent and exposure to X-ray film.

### 4.8. Statistical Analysis

Data are presented as mean ± standard deviation. Statistical analyses were performed using one-way analysis of variance. Data were compared using the Student’s *t* test. The level of statistical significance was set at * *p* < 0.05, ** *p* < 0.01, and *** *p* < 0.001.

### 4.9. Predict the Potential Mechanism of HDAC8 Inhibitor through Multi-Database Platform

Indirect process: CLUE calculated the connectivity score of shRNA HDAC8 among their one million profiles and ordered the similarity to opposite compounds as well as gene perturbations. The criteria were filtered above 90 positive scores and collected at each instance targeted genes as shRNA HDAC8 regulators in biological function. This genes list was input into the CPDB platform for enrichment analysis to obtain clear pathways information. Direct process: Based on BMX-L1000 gene expression data, both up-regulation and down-regulation gene list respond to the drug biological function in HepG2 cells. BMX (1 μM) was compared to DMSO control to define the significant differential expression genes (DEGs) according to ±1.5 fold change, *p*-value < 0.05. Hence, the DEGs were used as inputs to query CPDB for pathway analysis. To narrow down the prioritized pathways, we were interested in two results and selected the common element.

## 5. Conclusions

High HDAC8 expression in human GBM tissues and GBM-R cell lines correlated with MGMT levels. The BMX and TMZ combination induced WT-p53 mediated apoptosis through WT-p53 mediated MGMT inhibition in GBM-R cell lines. Moreover, the BMX and TMZ combination also suppressed cell proliferation and GSC phenotype activity via the β-catenin/c-Myc/cyclin D1/SOX2 signaling pathway in GBM-R cell lines. Therefore, BMX could be a promising strategy for the precision personal treatment of WT-p53 and TMZ-resistant GBM patients.

## Figures and Tables

**Figure 1 ijms-22-05907-f001:**
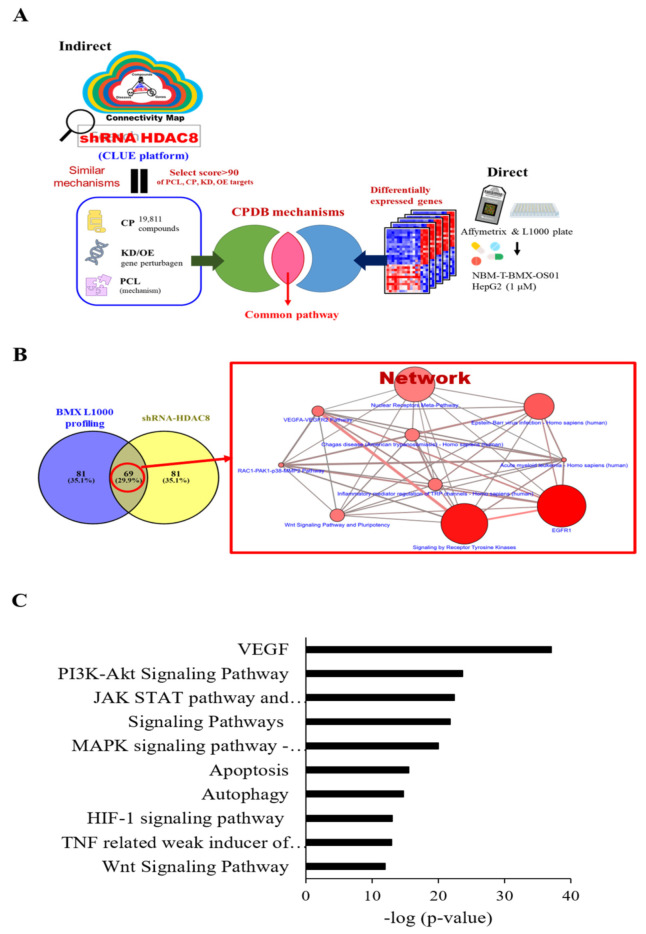
Pathway analysis for genes potentially associated with HDAC8 by bioinformatics tools. shRNA HDAC8 was entered into the CLUE database, and CP and PCL with a score of >90 were selected (**A**). The target genes were entered into the CPDB pathway analysis database (**B**) for further experiments. (**C**) Top 10 pathways for selecting CP and PCL (score > 90) for shRNA HDAC8. Ten pathways as below: VEGF; PI3K-Akt Signaling Pathway; JAK STAT pathway and regulation; Signaling Pathway; MAPK signaling pathway—Homo sapiens (human); Apoptosis; Autophagy; HIF-1 signaling pathway; TNF-related weak inducer of apoptosis (TWEAK) Signaling Pathway; Wnt Signaling Pathway. VEGF; PI3K-Akt Signaling Pathway; JAK STAT pathway and regulation; Signaling Pathway; MAPK signaling pathway—Homo sapiens (human); Apoptosis; Autophagy; HIF-1 signaling pathway; TNF-related weak inducer of apoptosis (TWEAK) Signaling Pathway; Wnt Signaling Pathway.

**Figure 2 ijms-22-05907-f002:**
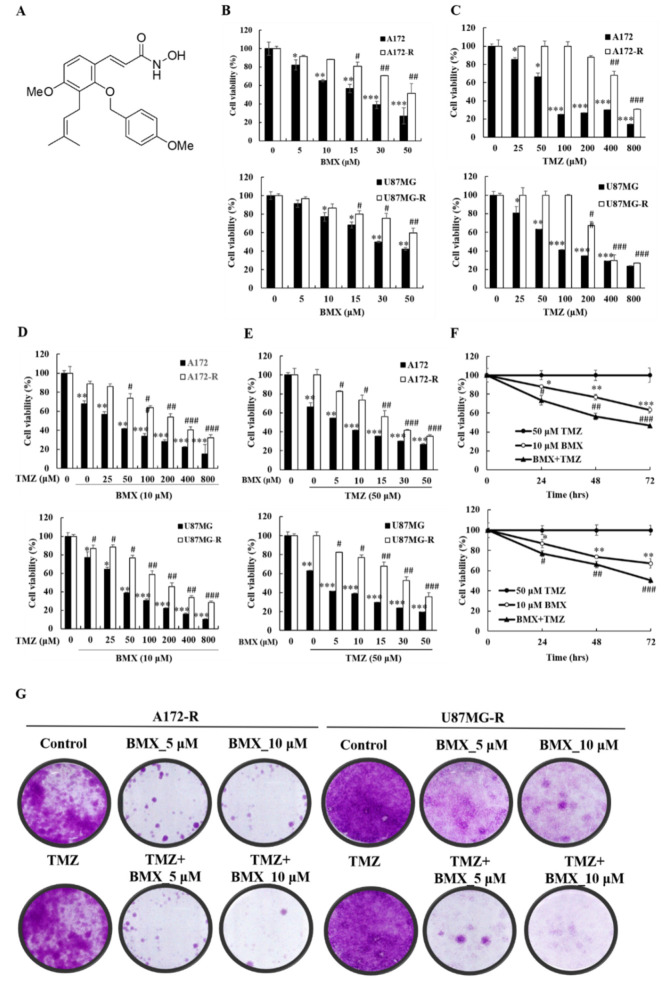
BMX inhibits the growth and proliferation of GBM cells (U87MG and A172) and the BMX and TMZ combination inhibits the growth and proliferation in GBM-R cells (U87MG-R and A172-R). (**A**) Chemical structure of BMX. (**B**) Cell viability of GBM and GBM-R cell lines after treatment with 0.5, 10, 15, 30, or 50 µM BMX. (**C**) Cell viability of GBM and GBM-R cell lines after treatment with 0.25, 50, 100, 200, 400, or 800 µM TMZ. (**D**) GBM and GBM-R cell viability after treatment with 10 µM BMX with or without TMZ at various concentrations (0.25, 50, 100, 200, 400, or 800 µM) for 24 h. (**E**) GBM and GBM-R cell viability after treatment with 50µM TMZ with or without BMX at various concentrations (0.5, 10, 15, 30, or 50 µM) for 24 h. (**F**) GBM-R cell viability after treatment with 50 µM TMZ with or without 10 µM BMX for 24, 48, and 72 h. (**G**) Colony formation assay of GBM and GBM-R cell lines with BMX (0, 5, or 10 µM) with or without TMZ (50 µM) for 14 days. Data are represented as means ± SEM from three experiments. * *p* < 0.05, ** *p* < 0.01, *** *p* < 0.05 vs. control (A172 and U87MG); ^#^
*p* < 0.05, ^##^ *p* < 0.01, ^###^ *p* < 0.05 vs. (A172-R and U87MG-R).

**Figure 3 ijms-22-05907-f003:**
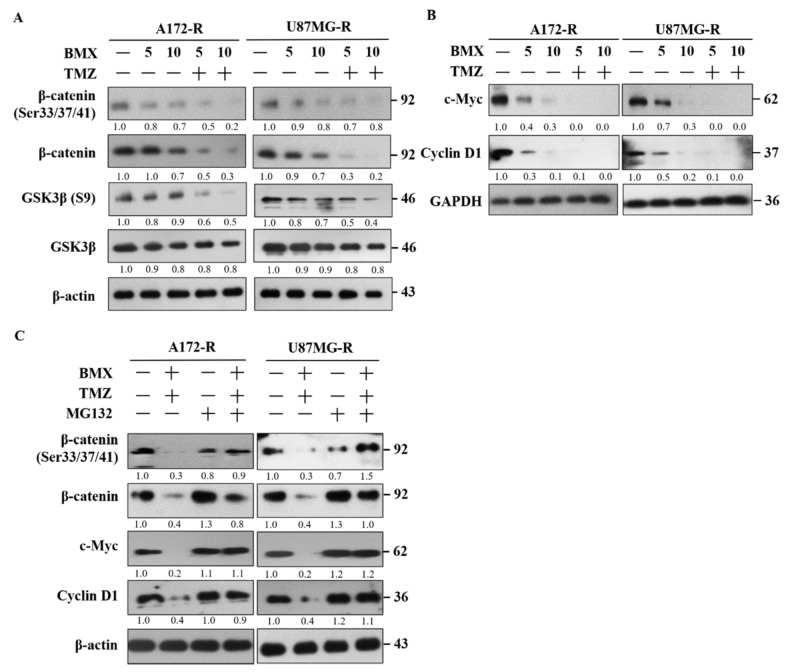
BMX enhanced the TMZ-mediated cytotoxic effect by targeting the Wnt/β-catenin/GSK3β pathway to suppress cell proliferation in GBM-R cells. (**A**) GSK-3β and β-catenin activation status of GBM-R cells after treatment with 5 or 10 µM BMX with or without 50 µM TMZ for 48 h. (**B**) c-Myc and cyclin D1 protein levels of GBM-R cells after treatment with 5 or 10 µM BMX with or without 50 µM TMZ for 48 h. (**C**) β-catenin (Ser33/37/41) phosphorylation status and changes in c-Myc and cyclin D1 protein expression in GBM-R cells after treatment with 10 µM BMX and 50 µm TMZ with or without 10 µM MG132.

**Figure 4 ijms-22-05907-f004:**
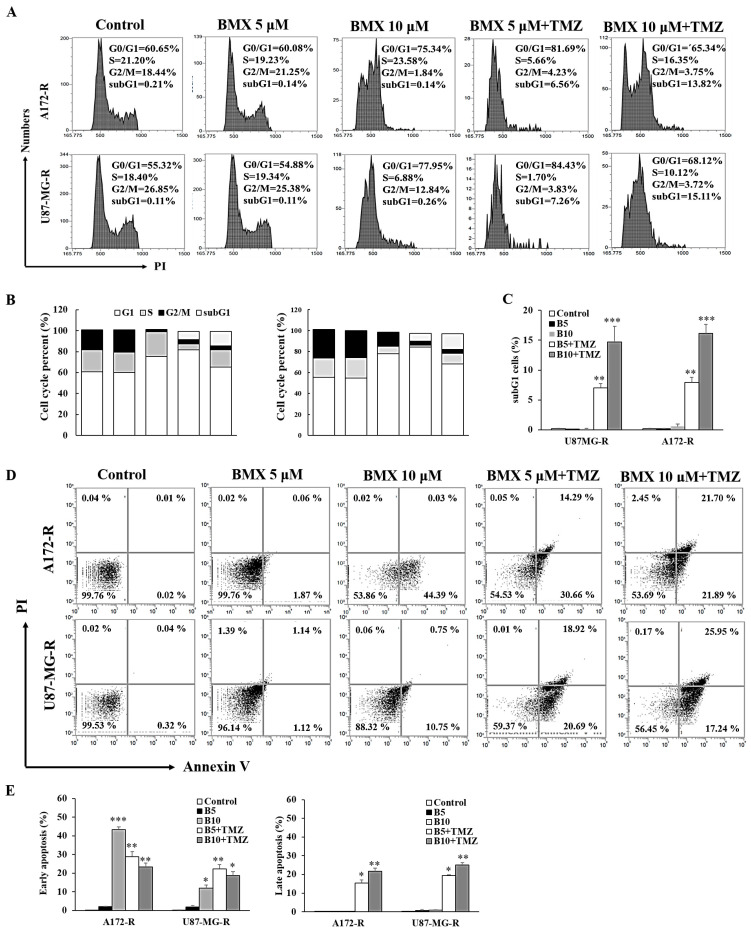
BMX and TMZ combination enhanced the TMZ-mediated cytotoxic effect by promoting TMZ-mediated apoptosis in GBM-R cells. (**A**) Cell cycle distribution of GBM (U87MG and A172) and GBM-R (U87MG-R and A172-R) cells treated with BMX for 48 h with or without a TMZ. (**B**) Percentages of cells in G0/G1, S, and G2/M phases are presented in the histograms. (**C**) Bar graph of the percentage of sub-G 1. (**D**) Annexin V/PI apoptosis assay of GBM (U87MG and A172) and GBM-R (U87MG-R and A172-R) cells treated with BMX for 48 h with or without TMZ. (**E**) Histograms illustrating the percentages of apoptotic cells. * *p* < 0.5, ** *p* < 0.01, *** *p* < 0.05.

**Figure 5 ijms-22-05907-f005:**
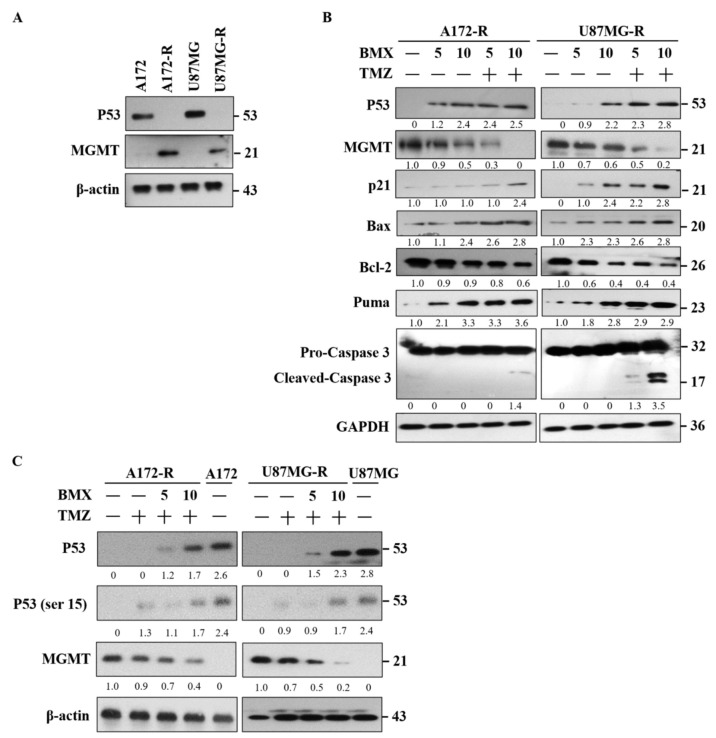
BMX and TMZ combination enhanced the TMZ-mediated cytotoxic effect by WT-p53 mediated MGMT inhibition in GBM-R cells. (**A**) The expression pattern of WT-p53 and MGMT on the GBM (U87MG and A172) and GBM-R (U87MG-R and A172-R) cell lines. (**B**) Protein alterations of WT-p53, MGMT, P21, Bax, Bcl-2, Puma, and cleaved caspase-3 after treatment with 5 or 10 µM BMX with or without 50 µM TMZ for 48 h on U87MG-R and A172-R cells. (**C**) Treatment of GBM (U87MG and A172) and GBM-R (U87MG-R and A172-R) cells with 5 or 10 µM BMX with or without 50 µM TMZ for 48 h reduced MGMT levels and increased WT-p53 and Phospho-WT-p53 levels (ser 15). β-actin was used as an internal control.

**Figure 6 ijms-22-05907-f006:**
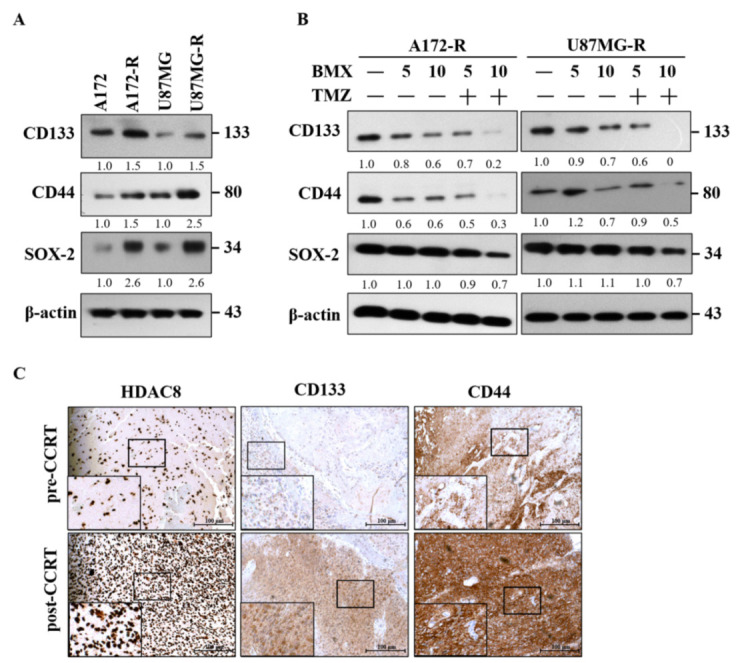
BMX and TMZ combination reduced GSC formation in GBM-R cells. (**A**) Status of CSC-related genes (CD133, CD44, and SOX2) expression between parental and resistant daughter cell lines. (**B**) Changes in CD133, CD44, and SOX2 protein levels after receiving 5 and 10 µM BMX with or without 50 µM TMZ for 48 h on U87MG-R and A172-R cells. (**C**) Immunohistochemical staining for HDAC8 and CSC-related genes (CD133 and CD44) in human primary GBM (the same patient before concomitant radiation and chemotherapy) and recurrent GBM tumor tissues (after concomitant radiation and chemotherapy) obtained through surgical biopsies.

**Figure 7 ijms-22-05907-f007:**
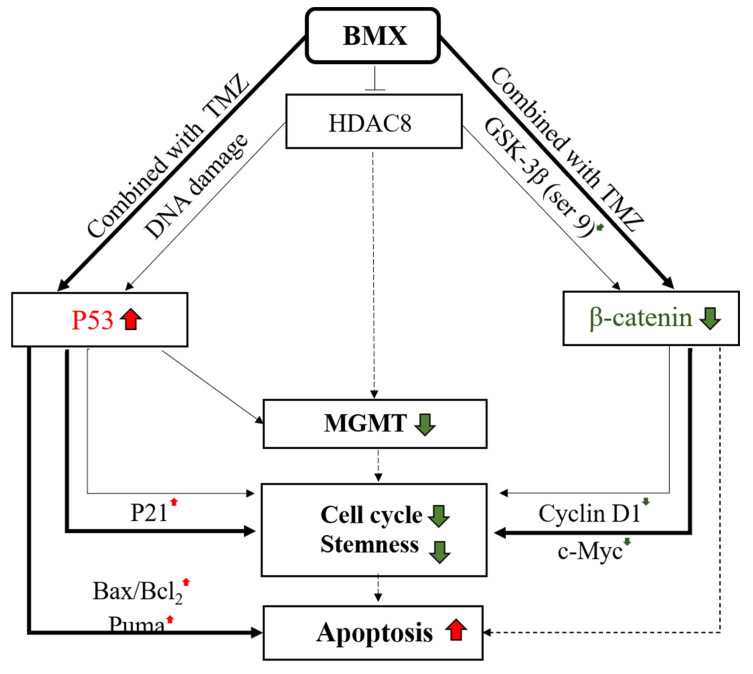
Working model of the mechanism of combination BMX with TMZ to overcome TMZ resistance in GBM-R cells. The model includes two main signaling pathways. The pathway on the right side: β-catenin/c-Myc/cyclinD1/Sox2 signaling pathway. When GBM-R cell lines were treated with BMX alone (thin lines) or BMX with TMZ (thick lines), GSK3β (S9) and active β-catenin decreased. The following c-Myc and cyclin D1 also decreased to induce cell cycle arrest and attenuate stemness activity, However, only BMX with TMZ (dashed lines) possibly induce apoptosis. The pathway on the left side: WT-p53 mediated MGMT inhibition. When GBM-R cell lines were treated with BMX alone or BMX with TMZ, WT-p53 increased and downregulated MGMT levels, cell cycle arrest, and stemness in both BMX alone (thin lines) and BMX with TMZ (thick lines). However, WT-p53 and DNA damage marker (WT-p53-ser15, not shown) increased the following the activation of cell cycle arrest marker (P21) and proapoptotic markers (BAX/Bcl2, and Puma) to induce apoptosis and cell death only in BMX with TMZ (thick lines), indicating profound DNA damage. Color red indicated up-regulation. Color green indicated down-regulation.

## Data Availability

Data is contained within the article or Appendix A.

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
