# Peer review of "NBM-BMX, an HDAC8 Inhibitor, Overcomes Temozolomide Resistance in Glioblastoma Multiforme by Downregulating the β-Catenin/c-Myc/SOX2 Pathway and Upregulating p53-Mediated MGMT Inhibition"

_ijms, 2021, doi:10.3390/ijms22115907_

Round 1

Reviewer 1 Report

Authors have investigated different human cell lines (sensitive and resistant to TMZ), regarding the potential beneficial effects of a concomitant treatment with HDAC inhibitors (BMX in this work). Joint treatment proved able to induce cell cycle arrest and apoptosis, as well as GSC markers. Results point to a possible help in treatment of TMZ resistant cases.

Major points

1.Authors need to revise English language again. There are points with confusing sentences, or probably missing parts of the sentence. I will try to list some, but a more extensive language review is needed for the sake of clarity.

2.I suggest to join the supplementary figures and tables in a single document. Please add captions for all supplementary figures and tables. Still, all supplementary figures and tables should be cited in the main text.

3.Line 144: if authors want to use the word “synergism”, a methodological and numerical approach to calculate should be used, which was not the case in this paper. Example: Chou-Talalay method (https://cancerres.aacrjournals.org/content/70/2/440).

4.Line 153, the IC50 value for TMZ in U87 seems significantly lower than values reported in literature. Although I understand that the core of the work is the joint treatment, authors should at least compare themselves with other authors to confirm the robustness of the approach used for cell viability assessment.

5.Regarding flow cytometry results, TMZ alone is not shown so it is difficult to infer the advantages of joint treatment in this case. Was it done? What was the result?

6.Figure 7: I understand and appreciate the effort of doing a summary figure including the mentioned pathways. However, for me it is not clear at all. I missed some indication in the squares of cell cycle and apoptosis if it is supposed to increase or decrease with or without treatment, not easily deduced in the actual figure. Please try a more clear summary.

  1. Authors should recognize that methods in vitro may not fully mimic the in vivo situation. For example, the half life of TMZ in vivo is really short and only a fraction of the administered TMZ arrives to tumours, as opposed to the in vitro situation in which a continuous exposition of 24 to 72h is proposed. I am assuming that no data yet about pharmacokinetics (even theoretical for example with e.g. http://www.swissadme.ch/). Since we are still at the in vitro point, ok, but for TMZ it is already known. Then, only a word of caution that there is still way to go.

Minor points

1.Abbreviations need to be defined the first time they appear. This is correctly done most times, but for example in lines 116-118 it is inverted (first abbreviation, later the definition).

2.Please note that in introduction, sometimes the HDACis which should appear together as defined in line 65, is appearing separate that made sentence construction really weird.

3.Line 81, “HDACi have been used in many clinical trials for cancer treatments” can you put some examples of clinical trial numbers?

4.Lines 87 to 96 are results and does not have to appear in introduction

5.Line 103, define “PDB”

6.Line 110, “indirective approach” what does it mean exactly? Indirect approach?

7.Line 122 “highly potential pathways” is a weird construction, please rephrase

8.Figure 1: Figure 1B is hardly seen, please try to enlarge letters in network part. Figure 1C, tags in “y” axis are not properly seen, as if they have been cut

9.Around line 146, please use fold change to explain cell viability decrease in the relevant time points.

10.Line 156: when working in vitro, we does not refer to treatments as “dose”, we use “concentration”.

11.Figure 2: In Figure 2F, treatment of resistant lines with TMZ gives always 100% viability, is this so? I would expect of course, and if they are resistant, small viability decreases but it does not seem to be the case. Can you explain it better? Is the viability decrease pure 0 even after 72h? In the caption of figure 2, please state for all instances the time of treatment, as it is done for 2D.

12.Figure 3: Typographic error in figure 3C “catanin”

13.Line 245-246  “By overall survival plot (supplementary figure 4)”, this figure is not a survival plot

14.Lines 260-262, authors mention a scatter plot, was this shown in some figure? Is this supplementary figure 5? All figures must be cited in main manuscript

15.Figure 6: the histopathological part is a bit confusing, because in a first moment, readers tend to think that is the same patient before and after treatment. This is properly explained in caption, but maybe some comment already in figure must be done. Please state also magnifications and the meaning of the squares shown

16.Discussion: Lines 299-301 this sentence is unreadable, I don’t even understand what do you mean here. Please rephrase

17.Lines 308-309: Sentence that starts by “First, the…” it is incomplete, words missing.

18.Methods: line 482. Is this “90 positive scores” a threshold established by authors, by literature?

Author Response

Responses to reviewer comments:

Response to reviewer 1:

Authors have investigated different human cell lines (sensitive and resistant to TMZ), regarding the potential beneficial effects of a concomitant treatment with HDAC inhibitors (BMX in this work). Joint treatment proved able to induce cell cycle arrest and apoptosis, as well as GSC markers. Results point to a possible help in treatment of TMZ resistant cases.

To answer point-by-point the details of the revisions to reviewer 1:

Major points

Q 1. Authors need to revise English language again. There are points with confusing sentences, or probably missing parts of the sentence. I will try to list some, but a more extensive language review is needed for the sake of clarity.

Ans: Thanks for your comments. We have already made adequate modification

Q 2. I suggest to join the supplementary figures and tables in a single document. Please add captions for all supplementary figures and tables. Still, all supplementary figures and tables should be cited in the main text.

Ans: Thanks for your comment. We added captions for all supplementary figures and tables. We also moved Supplementary Figure 5 as Supplementary Figure 1B and revised the legends of Supplementary Figure 1A and 1B.

Q 3. Line 144: if authors want to use the word “synergism”, a methodological and numerical approach to calculate should be used, which was not the case in this paper. Example: Chou-Talalay method (https://cancerres.aacrjournals.org/content/70/2/440).

Ans: Sorry for our mistake. In this study, our hypothesis was based on drug combination, but not “synergism” for both drugs effects. Therefore, we change it as “combination” instead of “synergism”. We also made modification in lines 43, 145 and 380

Q 4. Line 153, the IC50 value for TMZ in U87 seems significantly lower than values reported in literature. Although I understand that the core of the work is the joint treatment, authors should at least compare themselves with other authors to confirm the robustness of the approach used for cell viability assessment.

Ans: IC50s of TMZ-sensitive cell lines, such as A172, U87 and U251 were varied from 7-250 μM. In this study, our data showed IC50 values of TMZ alone were 73.48±3.65 μM/80.99±1.68. It appears our data is acceptable for reviewing literatures. Please see reference: Temozolomide resistance in glioblastoma multiforme. Genes & Diseases (2016) 3,198-210

Q 5. Regarding flow cytometry results, TMZ alone is not shown so it is difficult to infer the advantages of joint treatment in this case. Was it done? What was the result?

Ans: Your question is well taken. We also performed flow cytometry for TMZ alone. The result revealed cell cycle is equal to control group, less than BMX alone or combination group. Therefore, we though combination group had advantages, comparing BMX alone and TMZ alone (We did not put into the main figure)

Q 6. Figure 7: I understand and appreciate the effort of doing a summary figure including the mentioned pathways. However, for me it is not clear at all. I missed some indication in the squares of cell cycle and apoptosis if it is supposed to increase or decrease with or without treatment, not easily deduced in the actual figure. Please try a more clear summary.

Ans: Thanks for your comment. We slightly modified our Figure 7 as new Figure 7.

Q.7 Authors should recognize that methods in vitro may not fully mimic the in vivo situation. For example, the half life of TMZ in vivo is really short and only a fraction of the administered TMZ arrives to tumours, as opposed to the in vitro situation in which a continuous exposition of 24 to 72h is proposed. I am assuming that no data yet about pharmacokinetics (even theoretical for example with e.g. http://www.swissadme.ch/). Since we are still at the in vitro point, ok, but for TMZ it is already known. Then, only a word of caution that there is still way to go.

 Ans: Your comment is well taken. We agree and recognize the fact what you mentioned about TMZ. Thanks for your reminding. 

Minor points

Q 1. Abbreviations need to be defined the first time they appear. This is correctly done most times, but for example in lines 116-118 it is inverted (first abbreviation, later the definition).

Ans: Thanks for your comment. We already corrected it in lines 116-118

Q 2. Please note that in introduction, sometimes the HDACis which should appear together as defined in line 65, is appearing separate that made sentence construction really weird.

Ans: Thanks for your comment. We already corrected it in lines 80, 81 and 83

Q 3. Line 81, “HDACi have been used in many clinical trials for cancer treatments” can you put some examples of clinical trial numbers?

Ans: Thanks for your question.

We list the result of HDACi in clinical trials: complete:372, Active: 71, Such as :

Safety,Tolerability and MTD KA2507 (HDAC6 Inhibitor) in Patients With Solid Tumours (HDAC6i) ClinicalTrials.gov Identifier: NCT03008018

Investigation of the HDAC4 Copy Number Variation and Its Effect on Gene and Protein Expression in Patients With ASD ClinicalTrials.gov Identifier: NCT03670381

Q 4. Lines 87 to 96 are results and does not have to appear in introduction

Ans: We agree it and modified it in lines 87-90

Q 5. Line 103, define “PDB”

Ans: We defined CPDB as “ConsensusPathDB” in line 97 (http://cpdb.molgen.mpg.de/)

Q 6. Line 110, “indirective approach” what does it mean exactly? Indirect approach?

Ans: indirective approach mean that we used bioinformatics tools to approach for HDAC8 inhibition function on GBM cell lines, but not direct approach by experiments

Q 7. Line 122 “highly potential pathways” is a weird construction, please rephrase

Ans: Thanks for your comment. We already corrected it as” possible potential pathways”

Q 8. Figure 1: Figure 1B is hardly seen, please try to enlarge letters in network part. Figure 1C, tags in “y” axis are not properly seen, as if they have been cut

Ans: We have already submitted Figure files by tiff. Therefore, we though the quality of Figure 1B is enough for enlarge to see the letters in network part. We also added the complete title of pathways in the legends of Figure 1C.

Q 9. Around line 146, please use fold change to explain cell viability decrease in the relevant time points.

Ans: Figure 2A-E are optimal concentrations for treatment. Only Figure 2F is the cell viability for drugs combination in our further experiments. Therefore, we added fold change of cell viability of Figure 2F as you suggested in lines 156-157 

Q 10. Line 156: when working in vitro, we does not refer to treatments as “dose”, we use “concentration”.

Ans: Thanks for your comment. We have already corrected it in line 152

Q 11. Figure 2: In Figure 2F, treatment of resistant lines with TMZ gives always 100% viability, is this so? I would expect of course, and if they are resistant, small viability decreases but it does not seem to be the case. Can you explain it better? Is the viability decrease pure 0 even after 72h? In the caption of figure 2, please state for all instances the time of treatment, as it is done for 2D.

Ans: We maintain TMZ-resistant cell lines in 50 μM TMZ and added 50μM TMZ for maintain concentration. Therefore, 100% viability in TMZ alone in Figure2F from 0 till 72 hrs is correct. However, figure 2D is combination BMX 10μM and TMZ 50μM in 24 hrs.

Q 12. Figure 3: Typographic error in figure 3C “catanin”

Ans: Sorry for our mistake. We have already corrected it.

Q 13 .Line 245-246  “By overall survival plot (supplementary figure 4)”, this figure is not a survival plot

Ans: Thanks for your comment. We changed it as colony formation assay in line 241

Q 14. Lines 260-262, authors mention a scatter plot, was this shown in some figure? Is this supplementary figure 5? All figures must be cited in main manuscript

Ans: sorry for our mistake, we added the supplementary S1B for scatter plot in line 256 and please also see major point Q2 answer above.

Q 15. Figure 6: the histopathological part is a bit confusing, because in a first moment, readers tend to think that is the same patient before and after treatment. This is properly explained in caption, but maybe some comment already in figure must be done. Please state also magnifications and the meaning of the squares shown

Ans: We agree your suggestion. We added the sentence “the same patient” in the legend of Figure 6C in line 290. The squares in Figure 6C is 100X magnifications and scale is also indicated 100μm.

Q 16. Discussion: Lines 299-301 this sentence is unreadable, I don’t even understand what do you mean here. Please rephrase

Ans: We corrected “HDAC is….. “ to “Although preclinical studies have indicated that HDACis have antitumor effects in glioma [28-30], their role in the treatment of chemotherapy-resistant GBM has still unclear” in line 295-296

Q 17. Lines 308-309: Sentence that starts by “First, the…” it is incomplete, words missing.

Ans: We eliminated “the”.

Q 18. Methods: line 482. Is this “90 positive scores” a threshold established by authors, by literature?

Ans:

For each query, we assessed whether it connected to its equivalent in Connectivity Map-L1000v1 at a high level of confidence (defined as nominal p (NP) ≤ 0.05, FDR ≤ 0.25 and |τ| ≥ 90).” τ is a connectivity map score.

Reviewer 2 Report

The authors statement the cotreatment with HDAC8 inhibitors (BMX) and temozolomide could overcome the GBM resistance cell line due to inhibit the cell viability, cell proliferation and induce cell cycle arrest and apoptosis. Overall, the manuscript is written very well with relevant background information and interesting LAB data.  I believe the results provide novel modality for the TMZ resistant GBM treatment. This study will thus offer new insights on improving the long-term survival rates of TMZ resistant GBM patients. The findings in this manuscript were innovative and will have an impact on the GBM field.

I suggest accepting the paper as it is. 

However, the issue need to be addressed.

Multiple GBM cell lines are known to contain TMZ resistant cells and several acquired TMZ resistant GBM cell lines have been developed for use in experiments designed to define the mechanism of TMZ resistance and
the testing of potential therapeutics. The authors use four GBM cell lines U87, U87R, A172, and A172R in this study. The U87R and A172R are adaptive TMZ resistant GBM cell lines.  Adaptive TMZ resistant GBM cells were presumed sensitive to treatment with a combination of the HDAC inhibitor and TMZ. However, the characteristics of intrinsic and adaptive TMZ resistant GBM cells have not been systemically compared. Maybe the authors can use intrinsic cell lines for the same study design in the future. (The authors do not have to perform the new intrinsic TMZ resistant GBM cell lines in this manuscript.)

Author Response

Responses to reviewer comments:

Response to reviewer 2:

The authors statement the cotreatment with HDAC8 inhibitors (BMX) and temozolomide could overcome the GBM resistance cell line due to inhibit the cell viability, cell proliferation and induce cell cycle arrest and apoptosis. Overall, the manuscript is written very well with relevant background information and interesting LAB data.  I believe the results provide novel modality for the TMZ resistant GBM treatment. This study will thus offer new insights on improving the long-term survival rates of TMZ resistant GBM patients. The findings in this manuscript were innovative and will have an impact on the GBM field.

I suggest accepting the paper as it is. 

However, the issue need to be addressed.

To answer point-by-point the details of the revisions to reviewer 2:

Multiple GBM cell lines are known to contain TMZ resistant cells and several acquired TMZ resistant GBM cell lines have been developed for use in experiments designed to define the mechanism of TMZ resistance and
the testing of potential therapeutics. The authors use four GBM cell lines U87, U87R, A172, and A172R in this study. The U87R and A172R are adaptive TMZ resistant GBM cell lines.  Adaptive TMZ resistant GBM cells were presumed sensitive to treatment with a combination of the HDAC inhibitor and TMZ. However, the characteristics of intrinsic and adaptive TMZ resistant GBM cells have not been systemically compared. Maybe the authors can use intrinsic cell lines for the same study design in the future. (The authors do not have to perform the new intrinsic TMZ resistant GBM cell lines in this manuscript.)

Ans: Thanks for your appreciation. Your question is well taken and we will perform new experiments in the future.

Round 2

Reviewer 1 Report

Paper and associated supplementary file have significantly improved.

I have detected 2 typos, please correct:

  • Line 320 of the main manuscript (pdf): IS, not HAS
  • Caption in figure 1 of the supplementary material: MUTANT not MUTATNT